# Comparative genomic analysis of thermophilic fungi reveals convergent evolutionary adaptations and gene losses
Andrei S. Steindorff [1], Maria Victoria Aguilar-Pontes [2,16], Aaron J. Robinson [3], Bill Andreopoulos[1], Kurt LaButti [1], Alan Kuo [1], Stephen Mondo [1], Robert Riley [1], Robert Otillar[1], Sajeet Haridas [1], Anna Lipzen [1], Jane Grimwood [1,4], Jeremy Schmutz [1,4], Alicia Clum[1,5], Ian D. Reid[2], Marie-Claude Moisan[2], Gregory Butler[2], Thi Truc Minh Nguyen[2], Ken Dewar[6], Gavin Conant [7], Elodie Drula[8], Bernard Henrissat [9], Colleen Hansel [10], Steven Singer[11], Miriam I. Hutchinson[12], Ronald P. de Vries [13], Donald O. Natvig[12], Amy J. Powell[14], Adrian Tsang[2] & Igor V. Grigoriev [1,15] ✉

Thermophily is a trait scattered across the fungal tree of life, with its highest prevalence within three fungal families (Chaetomiaceae, Thermoascaceae, and Trichocomaceae), as well as some members of the phylum Mucoromycota. We examined 37 thermophilic and thermotolerant species and 42 mesophilic species for this study and identified thermophily as the ancestral state of all three prominent families of thermophilic fungi. Thermophilic fungal genomes were found to encode various thermostable enzymes, including carbohydrate-active enzymes such as endoxylanases, which are useful for many industrial applications. At the same time, the overall gene counts, especially in gene families responsible for microbial defense such as secondary metabolism, are reduced in thermophiles compared to mesophiles. We also found a reduction in the core genome size of thermophiles in both the Chaetomiaceae family and the Eurotiomycetes class. The Gene Ontology terms lost in thermophilic fungi include primary metabolism, transporters, UV response, and O-methyltransferases. Comparative genomics analysis also revealed higher GC content in the third base of codons (GC3) and a lower effective number of codons in fungal thermophiles than in both thermotolerant and mesophilic fungi. Furthermore, using the Support Vector Machine classifier, we identified several Pfam domains capable of discriminating between genomes of thermophiles and mesophiles with 94% accuracy. Using AlphaFold2 to predict protein structures of endoxylanases (GH10), we built a similarity network based on the structures. We found that the number of disulfide bonds appears important for protein structure, and the network clusters based on protein structures correlate with the optimal activity temperature. Thus, comparative genomics offers new insights into the biology, adaptation, and evolutionary history of thermophilic fungi while providing a parts list for bioengineering applications.

Thermophilic fungi have adapted to thrive in high-temperature environments. Despite a debate in the literature on how to classify thermophilic fungi, they can be categorized based on their temperature requirements for growth and survival: thermophilic fungi thrive in high-temperature environments (>45 °C), mesophilic fungi prefer lower temperatures (20–34 °C), and thermotolerant fungi can survive at high temperatures (<45 °C) but do not necessarily grow optimally in these conditions[1,2]. The adaptations to survive high temperatures include slower metabolic rates, a high proportion of saturated fatty acids in their phospholipids, rapid turnover of enzymes, synthesis of heat shock proteins, and unique mechanisms for lipid storage[3,4]. These adaptations allow thermophilic fungi to endure temperatures lethal to most other organisms. The impressive resilience and enzymatic abilities of

thermophilic fungi therefore offer vast potential for applications in biotechnology, biofuel production, bioremediation, paper production, and therapeutic biomolecule production[5].

The heat shock response is a fundamental adaptation to elevated temperatures that plays a pivotal role in the survival of thermophilic fungi[4]. These fungi have evolved mechanisms of persistent thermotolerance, including membrane lipid composition alterations. They have a high proportion of phosphatidic acids in their membrane lipids and, in response to heat shock, exhibit a dynamic shift with a decrease in the levels of phosphatidylcholine and phosphatidylethanolamine. Additionally, trehalose, a key player in thermotolerance, experiences a decrease in concentration upon heat exposure[4,6]. These molecular adaptations form the foundation of their remarkable ability to withstand elevated temperatures.

The biotechnological allure of thermophilic fungi is further illuminated through genomic investigations. Berka and collaborators[7] conducted a comparative analysis of the genomes of two thermophilic fungi, unveiling their remarkable capacity to hydrolyze all major polysaccharides present in biomass. Maheshwari and collaborators[8] highlighted the profound capability of these fungi to degrade polysaccharide constituents of biomass, with their extracellular enzymes exhibiting temperature optima for activity that aligns closely with or surpassing the organism's optimum growth temperature. Such findings underscore the promise of thermophilic fungi in biomass utilization and bioenergy production. Furthermore, a deeper

understanding of the evolutionary context of thermophilic fungi has been provided by Morgenstern[2], revealing that known thermophilic fungi are primarily concentrated in the Sordariales, Eurotiales, and Onygenales orders within the Ascomycota phylum, with the Mucorales order possibly harboring additional thermophilic species at the basal lineage of fungi.

To harness the biotechnological potential of thermophilic fungi, we aimed to understand better their mechanisms of thermotolerance and enzymatic potential through comparative genomics. This study explores 29 thermophilic fungal genomes from three different orders (Sordariales, Eurotiales, and Mucorales), with a focus on genetic adaptations, such as the increase in the third base of codon (GC3), reduction in the effective number of codon (ENC) and core genes content. In addition, we used a machine learning approach to classify thermophilic/mesophilic fungi based on four gene families with high accuracy. We also suggest that protein structure prediction using AlphaFold2[9] can be used to find patterns in biotechnologically important features, such as the optimal temperature of enzymes.

## Results
### Phylogeny and genomic features
The dataset we used for this analysis comprises genomes of 29 thermophiles, 8 thermotolerant fungi, and 42 mesophiles, totaling 79 species from Ascomycota, Basidiomycota, and Mucoromycota phyla (Fig. 1a and Supplementary Data 1). We used Mesquite (http://www.mesquiteproject.org) to

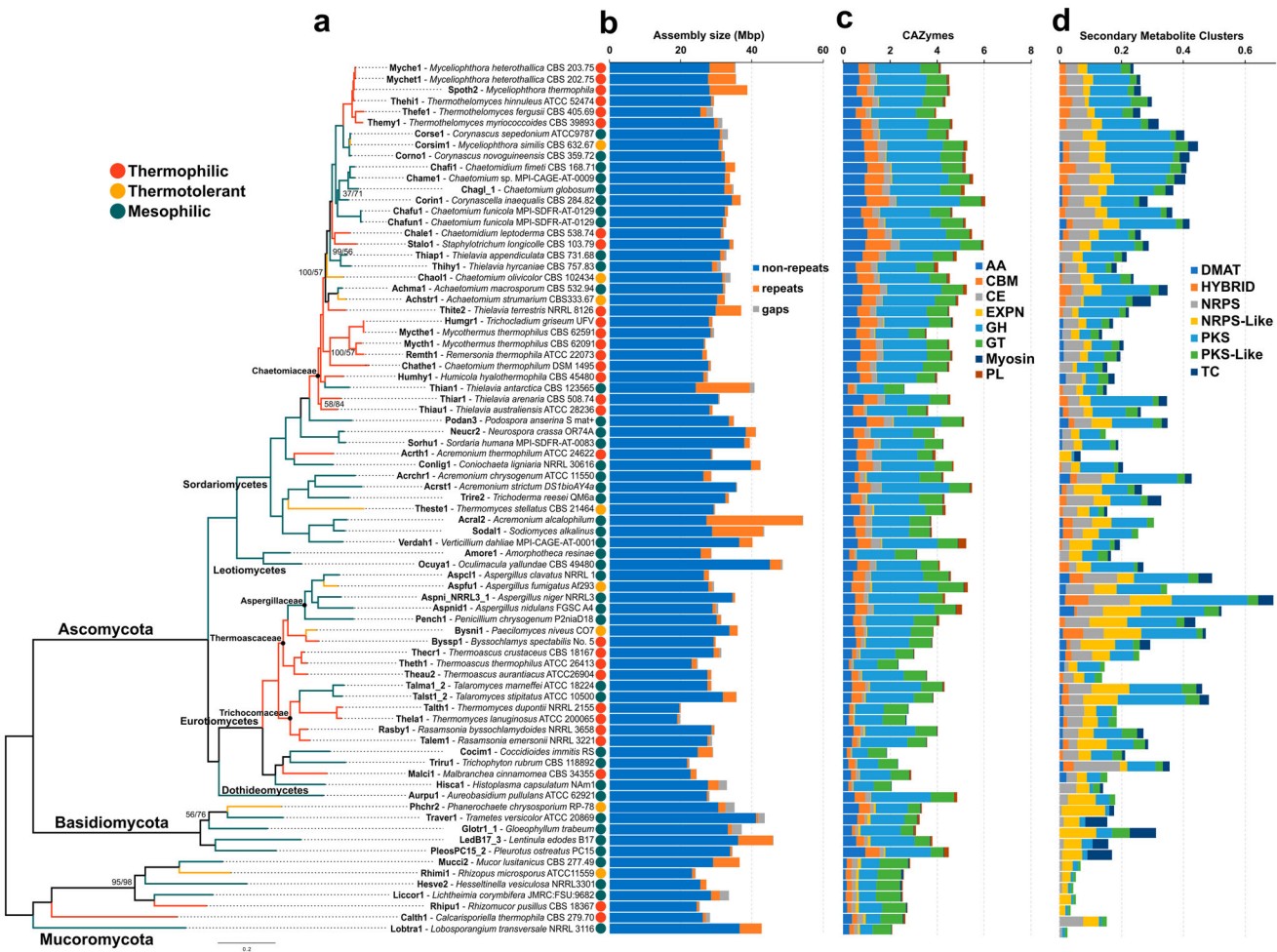

**Fig. 1 | Phylogeny and genome features of thermophilic, mesophilic, and thermotolerant fungi. a** Maximum likelihood tree of 79 fungi. The branch colors represent the ancestral ecological character states reconstructed with Mesquite using the Mk1 likelihood reconstruction model. **b** Assembly size in Mbp shows the distribution of repeats, gaps, and non-repeat content. **c** Normalized counts of CAZymes (Carbohydrate Active Enzymes) split into the classes AA (Auxiliary Activities), CBM Carbohydrate Binding Modules, CE (Carbohydrate Esterases), EXPN (Expansins), GH (Glycoside Hydrolases), GT (Glycoside Transferases, Myosin (Myosin Motor), and PL (Pectate Lyases). **d** Normalized counts of secondary metabolite clusters according to SMURF predictions.

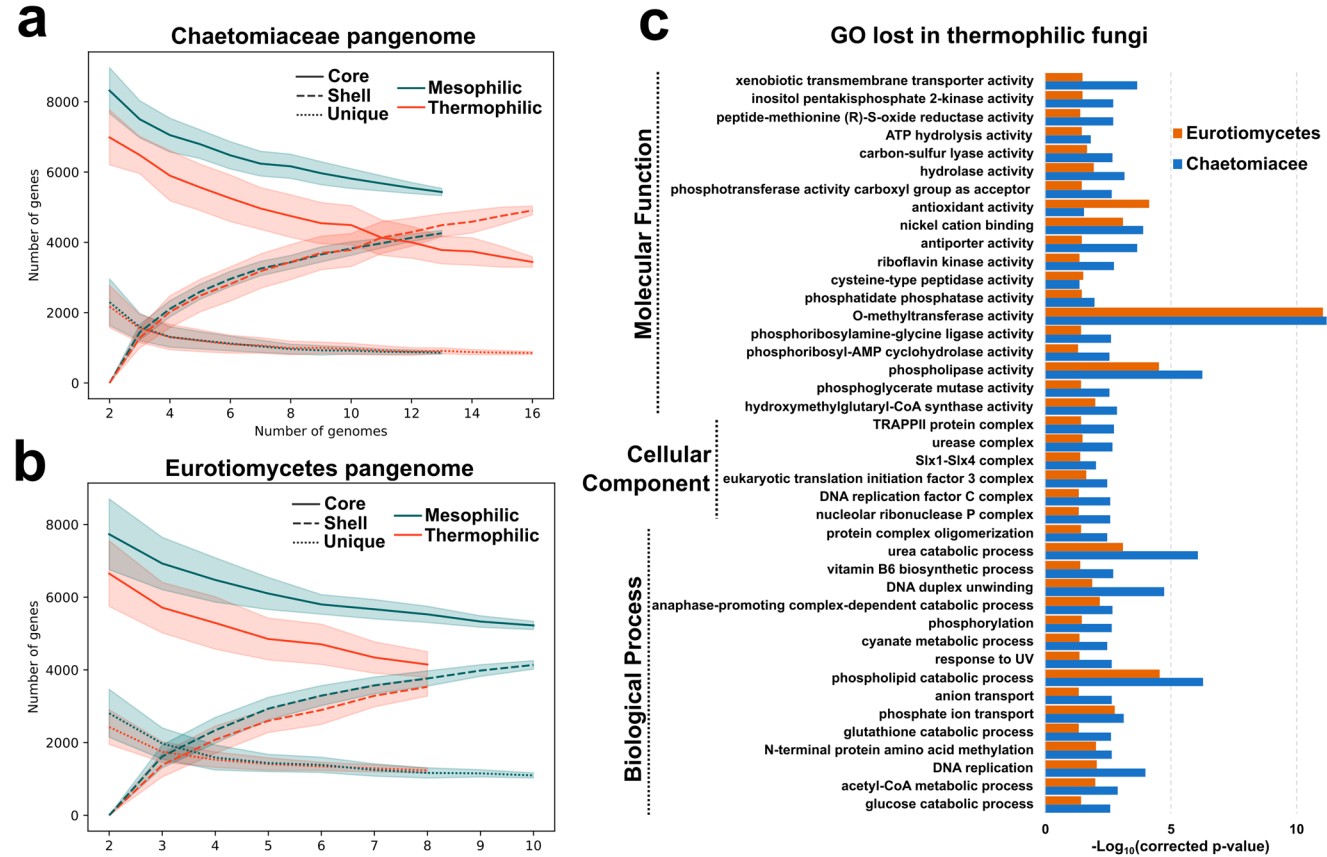

**Fig. 2 | Pangenome of Chaetomiaceae and Eurotiomycetes. a** Rarefaction curve of Chaetomiaceae pangenome split by Core/Shell/Unique and Mesophilic and Thermophilic. The bands represent the standard deviation of each average point. **b** Rarefaction curve of Eurotiomycetes pangenome split by Core/Shell/Unique in Mesophilic and Thermophilic. The bands represent the standard deviation of each average point in (**a, b**). **c** Enrichment analysis of GO terms lost in both thermophilic Chaetomiaceae and Eurotiomycetes core pangenomes compared to Mesophilic in the same clade. The X-axis represents the log10(corrected *p*-value) using the Benjamini-Hochberg procedure.

reconstruct the ancestral state of the clades to find the common ancestor of Chaetomiaceae, Thermoascaceae, and Trichocomaceae families all to be thermophilic. Interestingly, only the Chaetomiaceae revealed two or more thermophilic-mesophilic transitions, with ancestral clades within the families being thermophiles. Similar results were found by Hensen and colleagues[10], where variation in genome features stems primarily from within-family evolution.

We observed that the genome sizes in thermophilic fungi are smaller overall ($p = 3.3 \times 10^{-5}$ – two-tailed Wilcoxon rank sum test [WT]) and when compared with their mesophilic counterparts (most phylogenetically close in our dataset). The pair *Acremonium thermophillum* (28.89 Mbp) and *Coniochaeta lignaria* (42.38 Mbp) is a good example, with the genome of thermophile 13.49 Mbp shorter than its counterpart. (Fig. 1b). Similar trends can be observed in *Talaromyces* and *Thermomyces*, with average thermophilic genomes reduced by 38.12%. The pattern is less clear in cases like *Rasamsonia*, which have genomes closer in size to *Talaromyces*, and the Mucoromycota phylum (Fig. 2b).

Thermophilic fungi are known to secrete a wide range of thermostable enzymes, in particular CAZymes[7] (Fig. 1c, and Supplementary Data 2). Among the classes of CAZymes within Chaetomiaceae, the auxiliary activities (AA) ($p = 0.047$ - WT), carbohydrate esterases (CE) ($p = 0.042$), and pectate lyases (PL) ($p = 0.001$ - WT) were significantly contracted in thermophiles. In Eurotiomycetes, in contrast, carbohydrate binding modules (CBM - WT) and expansins (EXPN - WT) (both with $p = 0.039$) were significantly reduced. When the human pathogens (Triru1, Hisca1, and Cocim1) were excluded from the analysis, glycoside hydrolases (GH) were shown to be significantly reduced in thermophiles as well ($p = 0.0015$ - WT).

This is because human pathogen genomes encode a reduced number of GH compared to other mesophilic fungi ($p = 0.023$ - WT) (Fig. 1c).

## Thermophiles have a reduced core genome content

To further investigate the reduced genome size and, consequently, gene content in thermophilic fungi, we performed a pan-genome analysis with species belonging to the Chaetomiaceae family and Eurotiomycetes class in parallel. Interestingly, the rarefaction curve of unique and shell genes has a similar distribution in thermophiles and mesophiles. However, the core gene set of thermophiles is strikingly reduced in both clades, revealing this conserved pattern even in distantly related groups (Fig. 2a, b). Next, we investigated the GO terms significantly reduced in thermophiles from both clades (Fig. 2c). The most significant term reduced in thermophiles was O-methyltransferase activity, followed by phospholipase activity and phospholipid catabolic process. Several terms involved in primary metabolism (i.e., hydrolase activity, acyl-CoA metabolic process, and glucose catabolic process) and defense mechanisms were also contracted (i.e., xenobiotic transmembrane transporter activity, response to UV (Ultra-Violet)).

In addition, we investigated the expansion and contraction of gene families (HOGs) using the program CAFE v4.2.1[11]. In this analysis (Supplementary Data 3A), we found that most gene losses happened more recently, in phylogenetic tree nodes closer to the leaves instead of older nodes. For example, in the nodes representing the last common ancestor of the families Chaetomiaceae, Thermoascaceae, and Thichocomaceae, the net gain of gene families is negative, but close to zero. The overall average gene gains and losses show a clear trend of fewer genes gained by thermophilic

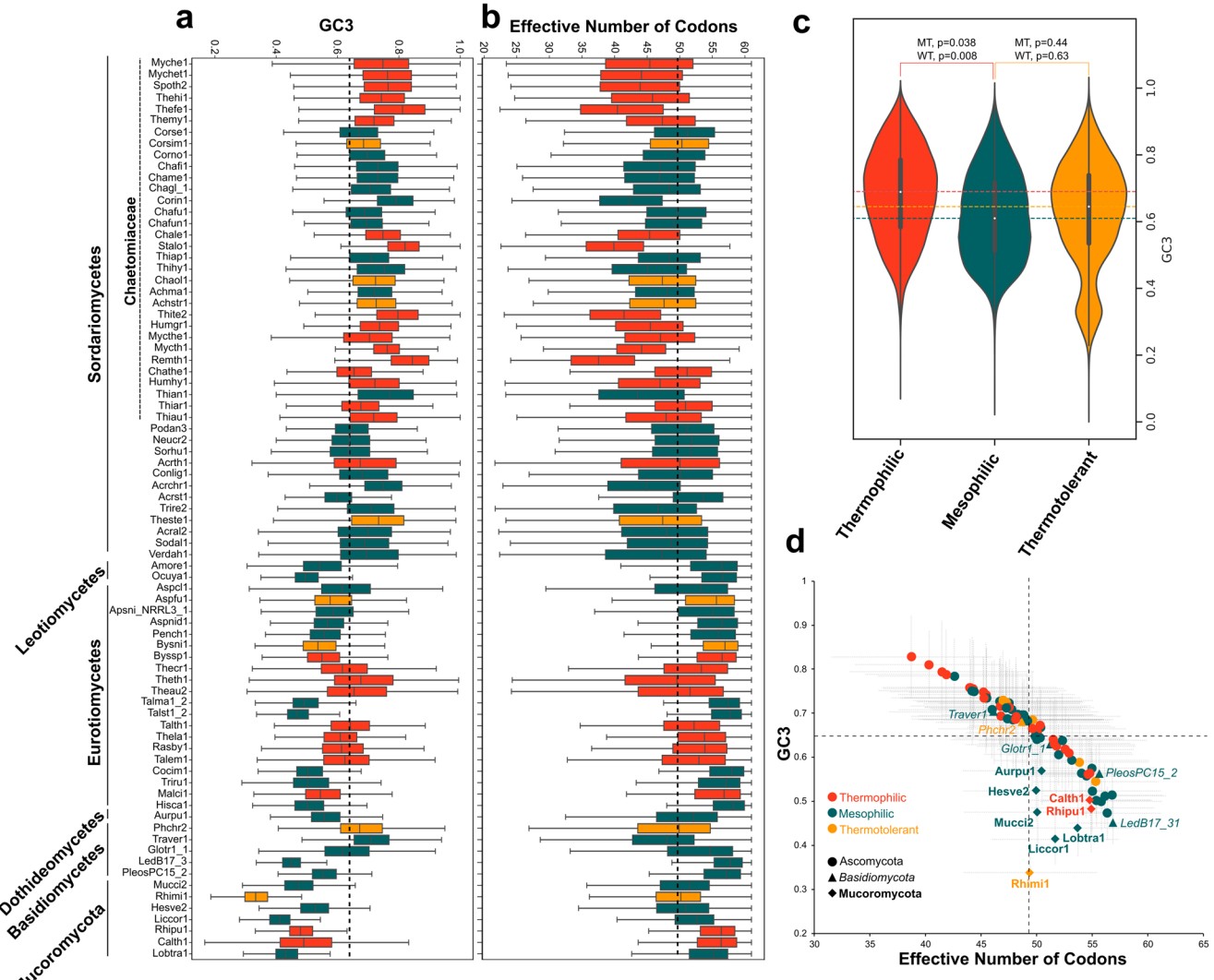

**Fig. 3 | GC content of third base of the codon (GC3) and Effective number of codons (ENC) in thermophilic fungi. a** GC3 distribution across 79 genomes. **b** ENC distribution across 79 genomes. **c** Violin plot of GC3 distribution in thermophilic, mesophilic, and thermotolerant genomes showing a significant difference between thermophilic and mesophilic using Moods median test (MT) and Wilcoxon rank sum test (WT). Horizontal lines show the median extended across the plot. **d** Scatterplots of average GC3 and ENC (gray dashed lines represent standard deviation) show a high correlation ($R^2 = -0.72$) between them. The black dashed lines represent the average GC3 and ENC of all genomes used in this study.

fungi (2.23 times fewer genes gained than mesophilic, $p = 0.00091$ - [WT]), and more genes lost (1.06 times more genes lost than mesophilic, $p = 0.6361$ - [WT]); and the opposite trend in mesophilic fungi. Interestingly, on average, thermotolerant fungi have a high number of gene gains and losses (Supplementary Data 3A).

### The effective number of codons and GC3 are altered in thermophilic fungi

Berka and collaborators[7] found an increase in GC3 in *Myceliophthora thermophila* and *Thileavia terrestris* transcripts, which to this day has not been confirmed to be true in other thermophilic fungi. In this study, we evaluated the coding sequences of 29 thermophiles (Fig. 3a) to show that even though GC3 was not increased for all of them, it is significant in the overall distribution (Fig. 3c). We also measured the degree of codon usage bias by computing the effective number of codons (ENC) for each species[12]. ENC ranges from 20 to 61, where 20 represents an extreme bias of using only one codon per amino acid, whereas 61 represents uniform synonymous codon usage, i.e., no bias[13]. The mean ENC values in thermophiles were 47.9, compared to 50.48 for mesophiles and 49.78 for thermotolerant. We found a higher fluctuation in ENC and GC3 along the Mucoromycota and

Basidiomycota branches ($R^2 = 0.11$, $p = 0.284$ versus $R^2 = 0.94$, $p < 1 \times 10^{-5}$ for Ascomycota) (Fig. 3d).

On the other hand, species in Ascomycota, in general, exhibit less variation in their codon bias than within clades such as Chaetomiaceae and Eurotiomycetes, where thermophilic fungi show significant biases compared to mesophilic counterparts ($p = 0.033$ and $p = 0.007$, respectively [WT]). Variations in the GC3 and ENC are anti-correlated ($R^2 = 0.72$, $p < 1 \times 10^{-5}$), with thermophilic fungi skewed toward the first quadrant (17 out of 29–58.6%) of the scatter plot (Fig. 3d), Basidiomycota split between first and fourth quadrant (2 and 3 species, respectively), and Mucoromycota all present in the fourth quadrant. In addition, early-diverging fungi exhibit, on average, less variation in ENC, but higher variation in GC3. Wint and collaborators[13] made a similar observation in their study with 450 fungal species.

### Machine learning can classify thermophilic and mesophilic fungi

We used Support Vector Machines (SVM) to identify hidden correlations between lifestyle and gene content, as described by Haridas and collaborators[14]. Large data sets are critical to machine learning, and only thermophilic and mesophilic rather than thermotolerant fungi had a

sufficient number of genomes in our dataset to be adequately separated using these methods. We selected a subset for training (20 thermophilic and 30 mesophilic fungi - Supplementary Data 3B) across the tree to identify genes that could discriminate between these two lifestyles. Using the HOGs dataset (ignoring unassigned genes), 100 orthogroups produced an accuracy of 70% each in their ability to distinguish between thermophilic and mesophilic lifestyles. The best discriminator among the HOGs with an average prediction accuracy of 0.81 was HOG0006878 (Supplementary Data 3C), found in 36% of the thermophiles and 83% of mesophiles and containing Cytochrome c/c1 heme lyases (PF16815). We used all 100 clusters in all possible combinations to improve prediction accuracy. Our results suggest that combining eight clusters increased accuracy to 94% (Supplementary Data 3C and Fig. 4). We did not include thermotolerant in the training set, only in the test set. We found that all thermotolerant species were classified as mesophilic, suggesting their genomic adaptations resemble mesophilic species.

These eight HOGs comprise the following Pfam domains: Two ABC transporters (PF00005 and PF12848), a critical component in response to environmental stresses[15]; Two short-chain dehydrogenases (PF00106), involved in the metabolism of a wide range of compounds and stresses[15,16]; Protein HRI1 (PF16815), a protein of unknown function in *Saccharomyces cerevisiae*[15–17]; Adenosine deaminase (PF00962) - plays an important role in nitrogen metabolism and may have a critical function in the regulation of fatty acid synthesis[18]; SAD/SRA domain (PF02182) - involved in 5mC methylation[19]; and a Calcineurin-like phosphoesterase (PF00149), enzymes potentially involved in DNA repair, stress response, and other cellular functions[20]. It is important to note that all HOGs seem to be reduced in thermophilic genomes, except the ABC transporter (PF12848), which interacts with ribosomes and modulates translation elongation in bacteria[21].

### Protein structures embed optima temperature in endoxylanases

Enzymes are the workhorses of biotechnological processes, and their performance is linked to their three-dimensional structure. By deciphering the structural intricacies of enzymes involved in biomass degradation (i.e., cellulases and xylanases), researchers can strategically engineer or select enzymes that are stable and highly active at elevated temperatures, a key requirement for efficient biomass conversion in industrial applications[2,7]. The AlphaFold2 protein structure prediction technology[22] has emerged as a transformative tool with profound biotechnological significance in biomass degradation. By providing highly accurate predictions of protein structures, AlphaFold2 empowers researchers to unlock a deeper understanding of the intricate molecular machinery involved in enzymatic processes. This newfound precision is invaluable when identifying the optimal temperature, pH, and enzymatic activity conditions for biomass degradation[23]. We used AlphaFold2 to predict the GH10 endoxylanase protein structures. Since this family is highly sampled in the Protein Data Bank (PDB)[24], the predicted structures had predicted template modeling (pTM) > 0.59 (Supplementary Data 4).

We built a protein structure similarity network using 318 GH10 endoxylanases with Foldseek[25] from the 79 genomes used in this study (Fig. 5). It is important to note that all GH10 proteins fell into two orthogroups (N0.HOG0000305 and N0.HOG0011586 - Supplementary Data 3B) based on the OrthoFinder[26] clustering results, one large cluster with 307 proteins, and a small one with 10 proteins (cluster 9 – Fig. 5a, b). It suggests that this family is highly conserved sequence-wise, so differences might be present at the structural level. We found that the number of disulfide bonds was the same within each cluster and different between them (Fig. 5), modulating their stability and constraining their conformational dynamics[27]. In addition, proteins from both mesophilic and thermophilic fungi were clustered together, with clusters with three proteins scattered along the phylogenetic tree, and with two proteins surrounding cluster 8 (Figure a). Also, we found some examples of proteins with long branch lengths (high evolutionary rate) losing structural similarity with closely related proteins (i.e. 38518_Stalo1, 654_Theste1, and 6258_Chathe1 - Fig. 5a).

We mapped available data on the optimal temperature of 31 proteins (9.7% - Supplementary Data 4), onto the obtained structural network (Fig. 5). Most sequences with optima between 55–60 °C (77%) fell in the same structural cluster (Fig. 5). On the other hand, there was no clear distribution pattern for enzymes with optima <55 °C. One hypothesis is that enzyme activity at higher temperatures requires a tight structural conformation, reflected by the high structural similarity. It is important to note that different labs and hosts used for heterologous expression can generate divergent enzymatic optimal temperatures[28,29].

## Discussion

In this study, we conducted a comprehensive analysis of 79 fungal genomes from three phyla, Ascomycota, Basidiomycota, and Mucoromycota, to identify genomic features of thermophilic fungi. There is a long evolutionary history between these three clades, dating to 769 MYA (CI: 605.2–1049.0 MYA)[30], so finding patterns covering such diversity is a significant challenge. Regarding the phylogeny of thermophilic fungi, we expanded the analysis performed by Morgenstern[2] to include 29 thermophilic fungi, and reconstructed genome-level phylogenetic relationships. In addition, we pinpoint that the families Chaetomiaceae (Sordariomycetes), Thermoascaceae, and Trichocomaceae (Eurotiomycetes) showed two or more thermophilic species. On the other hand, in the families Cephalothecaceae (Sordariomycetes), Malbrancheaceae (Eurotiomycetes), Lichtheimiaceae, and Calcarisporiellaceae (Mucoromycota), we found only one thermophilic species (Fig. 1a). We found that the ancestral state in these families with two or more thermophilic fungi is thermophilic, similar to prokaryotes[31].

It has been reported that specialization can lead to a reduction in genome size and gene content, maintaining genes critical for survival and losing genes no longer necessary outside their constrained lifestyle[32,33]. Two studies, including thermophilic bacteria and archaea, have shown that isolates with high optimum growth temperatures often have small genomes[34,35]. We found such a reduction in thermophilic fungi, more specifically, a loss of a large number of core genes involved mainly in primary/secondary metabolism and defense mechanisms (Fig. 2). We hypothesize that adaptation to high temperatures (>45 °C) represents a type of specialization involving a narrowing of niche space such that thermophiles have a reduced range of substrates as well as a less complex environment with respect to competing fungi, bacteria, and micro-invertebrates, resulting in reduced competition for nutrients and reduced predation. Another reduced function in our analyses involved "response to UV", which might indicate that thermophilic fungi evolved within local environments with protective barriers against solar radiation - including natural compost, herbivore droppings, and below the surface of sun-heated decomposing organic matter.

Interestingly, our machine learning approach to distinguish between thermophilic/mesophilic lifestyles revealed eight gene clusters mainly lost in thermophilic fungi, except the ABC transporter (PF12848). This domain is not involved in transmembrane transport as other ABC domains, but are specialized translation factors that interact with ribosomes, contributing to diverse cellular processes[21]. The other domains are often related to environmental stress (PF00005, PF00106, PF00149), regulation of fatty acids biosynthesis (PF00962), methylation (PF02182), and DNA repair (PF00149), similar to the genome reduction analysis. This shows that specific losses are more relevant for the thermophilic lifestyle and were carried out over almost 800 million years of fungal evolution.

Another important feature in thermophilic fungi is the increased GC3 and a biased ENC. It is known that adaptive mechanisms are the driver of interspecies codon usage patterns in fungi, with adaptive codon usage evident at the genome-level[13]. A similar pattern of GC3 was found for pyrophilous genomes, known fire-responsive fungal colonizers of post-fire soil[36], compared to mesophilic fungi. Since proteins produced by thermophilic fungi are known to be more thermostable[7,8], this difference in GC3 and ENC might confer an adaptation to higher temperatures, enhancing translation efficiency and protein stability.

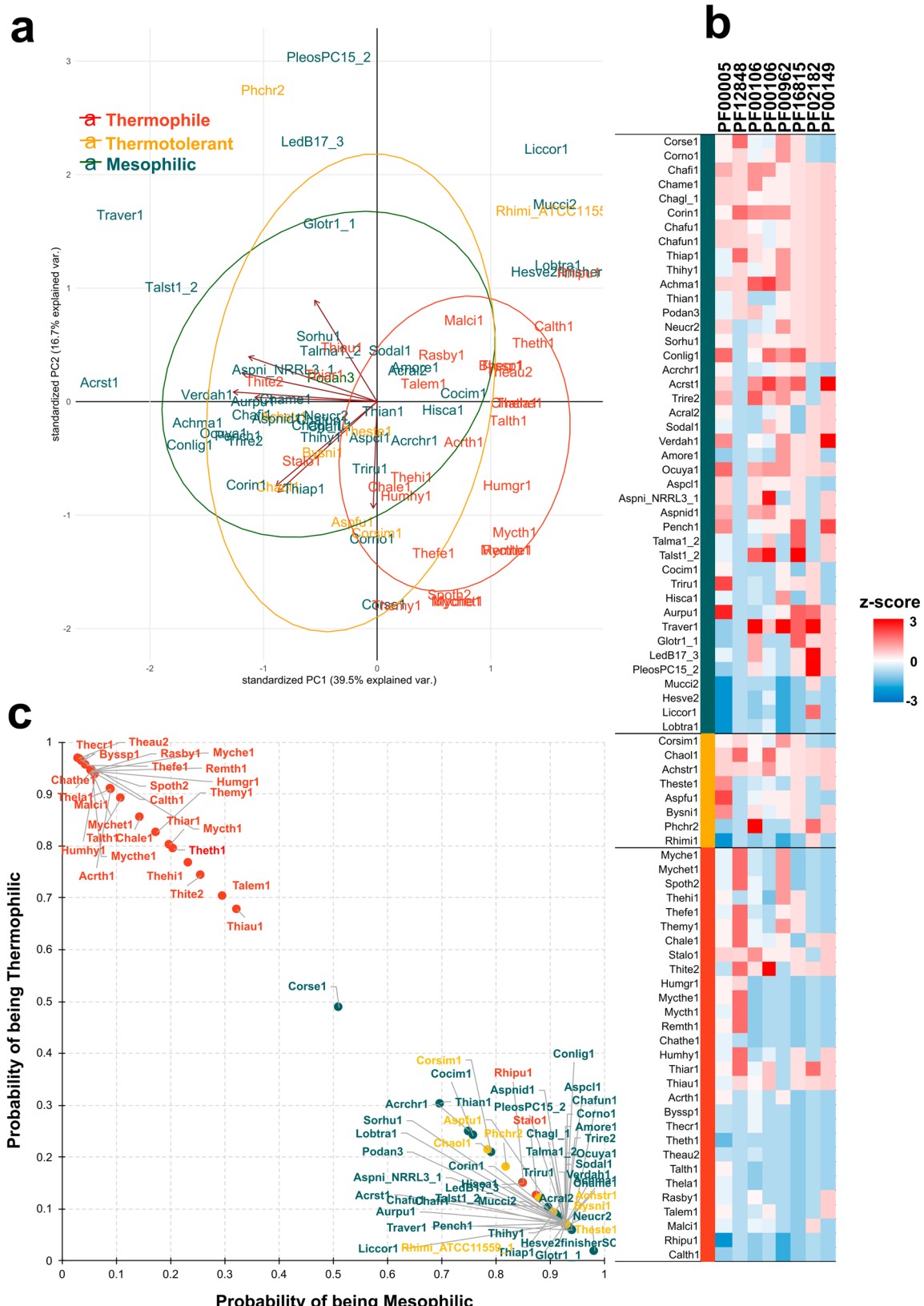

**Fig. 4 | Informative gene clusters to thermophilic lifestyle identified using support vector machine (SVM) classifier. a** Biplot PCA using the eight most informative clusters showing the separation between thermophiles and mesophiles. **b** Heatmap of the eight most informative HOGs showing prevalent gene loss in thermophiles. **c** Scatterplot of probabilities of a genome being a thermophile or mesophile based on these eight gene clusters containing the following Pfam domains: PF00005 and PF12848: ABC transporters; PF00106: short chain dehydrogenase; PF00962: Adenosine deaminase; PF02182: SAD/SRA domain (involved in 5mC methylation); PF16815: Protein HRI1; PF00149: Calcineurin-like phosphoesterase.

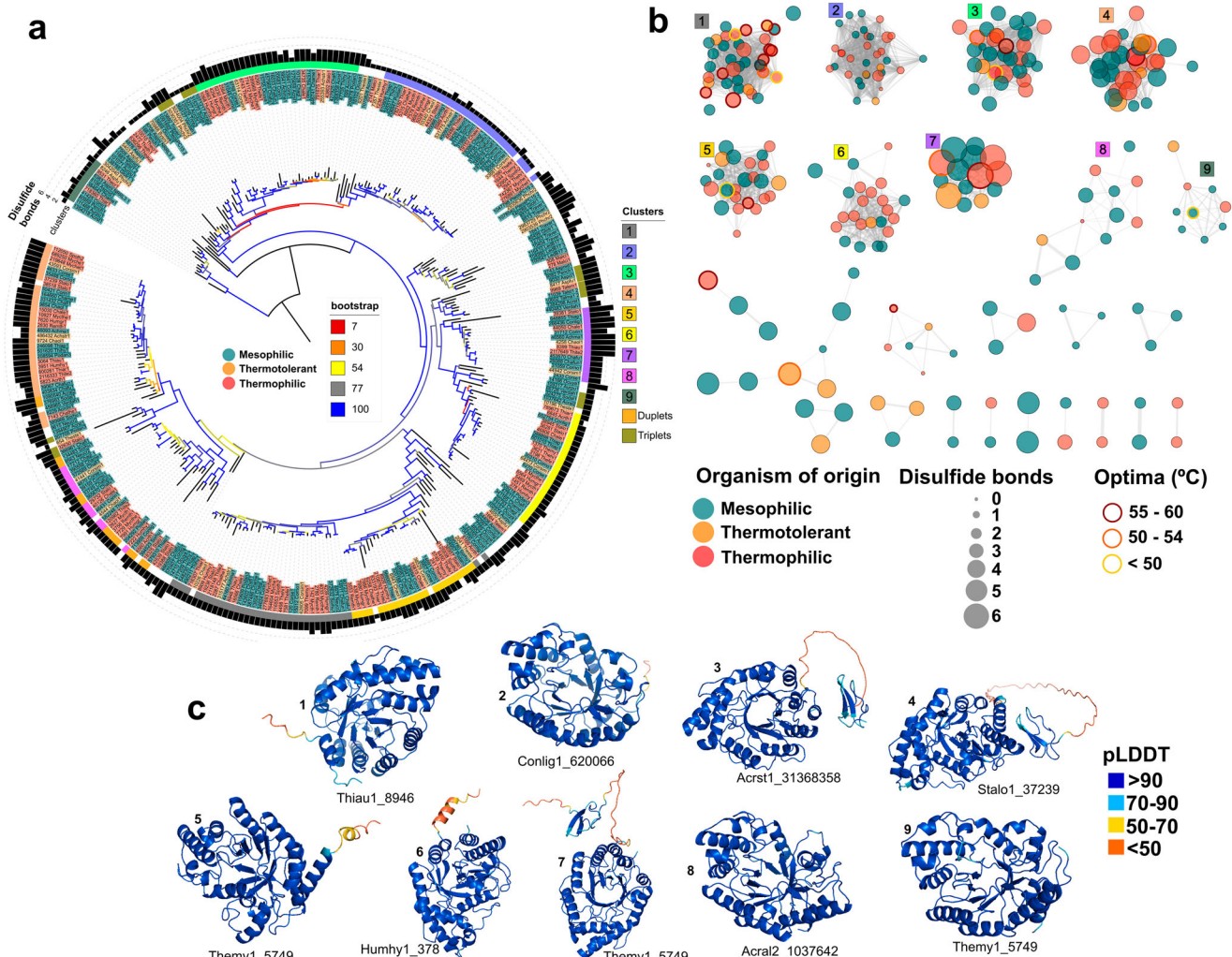

**Fig. 5 | Phylogeny and structural similarity network of GH10 endoxylanases using AlphaFold2. a** Phylogeny of GH10 endoxylanase built with iqtree [-m MFP -bb 10000 -safe]. The branches are colored based on the bootstrap value; the leaves' colors (green – mesophilic, orange – thermotolerant, red - thermophilic) represent the organism of origin; the color stripe in the outer circle represents the structural clusters shown in (**b**) (plus duplets - two protein clusters; and triplets - three protein clusters), and the bar chart (black) shows the number of disulfide bonds. **b** GH10 endoxylanase network clustered based on structural similarity using Foldseek. Edge thickness represents the percentage identity between protein structures. The clusters were separated based on percent identity (>0.7) and TMscore (>0.8) on a pairwise comparison of all endoxylanases using Foldseek. Colors (green – mesophilic, orange – thermotolerant, red - thermophilic) represent the organism of origin, and the size shows the number of disulfide bonds. Edges for pairs with percent identity <0.7 and TMscore < 0.8 were removed. **c** One structure from each cluster was randomly selected to show the prediction quality based on the AlphaFold2 pLDDT - a per-residue measure of local confidence scaled from 0 to 100.

The biotechnological utility of thermophilic fungi has been recognized for many years[5,7], and they are potential sources of enzymes with scientific and commercial interests, especially enzymes acting in lignocellulosic biomass. Optimal temperature, pH, and specific activity are the most important features of enzymes. With this in mind, we used a similarity network based on the AlphaFold2-based protein structure predictions for GH10 endoxylanases to find relevant information embedded in these structures (Fig. 5). We were able to show that within clusters of protein structures, the number of disulfide bonds may help define the optimal temperature. We found that proteins from both mesophilic and thermophilic fungi were present in the same clusters (Fig. 5b), which given the thermophilic lifestyle as the ancestral state of the fungal families used in this study (Fig. 1a), suggests that mesophilic fungi retained those thermophilic genes throughout their evolution. We also found that smaller structural clusters were scattered along the phylogenetic endoxylanase tree (cluster with three proteins), or close to each other (cluster with two proteins) suggesting phylogenetically close proteins can have relevant differences in the three-dimensional structure.

Since both mesophilic and thermophilic fungi can produce thermophilic enzymes, the protein structure combined with disulfide bonds can be used to predict the potential optimal temperature and other enzyme capacities based on their structural clustering. This approach not only enhances our understanding of the intricate relationships within thermophilic/mesophilic enzymes but also opens avenues for predictive tools in enzyme selection for heterologous expression experiments, thereby solidifying the pivotal role of thermophilic fungi in biotechnological applications.

The findings of this study provide valuable insights into the genomic features of thermophilic fungi and their specialized lifestyles. By identifying the genes that determine thermophiles using a machine learning approach, we can gain a better understanding of the factors that drive their evolution and ecological adaptations, as well as insights into the optimization of fungal-based industrial processes. Furthermore, the reduced genome and specialized lifestyle of these fungi make them excellent candidates for the study of stress responses, providing an opportunity to unravel the mechanisms underlying stress adaptation in

eukaryotic organisms. Overall, this study represents an important step toward the exploration of fungal diversity and genomic features and provides a solid foundation for future comparative genomics studies of thermophilic fungi.

## Methods

### Genome sequencing, assembly, and annotation

As described by Morgenstern and collaborators[2], the growth characteristics of 37 reported thermophilic or thermotolerant fungi, together with 42 mesophilic species, were examined at four temperatures: 22 °C, 34 °C, 45 °C, and 55 °C. Based on the relative growth performances, species with a faster growth rate at 45–55 °C than at 34 °C were classified as thermophilic, and species with better or equally good growth at 34 °C compared to 45 °C as thermotolerant[2]. All the genomes used in this study were collected from the MycoCosm[37] web portal. All metadata and GenBank/NCBI accessions are present in Supplementary Data 1. The 23 newly sequenced thermophilic strains in this study (Origin = CSFG (Centre for Structural and Functional Genomics – Concordia University) on Supplementary Data 1) were sequenced using Roche 454 pyrosequencing, Illumina, or Pacific Biosciences sequencing technologies. The Illumina assemblies were drafted from 2 x 150 bp, and 2 x 250 bp paired-end reads using SOAPdenovo[38]. PacBio sequence data were assembled with Masurca[39]. Pilon[40] was used to improve assemblies with a large number of contigs. CSFG Genomes were annotated with SnowyOwl[41] using transcriptomes downloaded from NCBI when available. Functional annotation for all 23 genomes was performed using the MycoCosm annotation pipeline[37]. For CAZy annotation (Supplementary Data 2), module composition and family assignment of all carbohydrate-active enzymes (CAZy) were performed just as for the daily updates of the CAZy database (http://www.cazy.org)[42]. The genome completeness was assessed with BUSCO[43] v5.2.2 [-m prot] using fungi_odb10 database (Supplementary Data 1).

### Phylogenetic analysis

The phylogenetic tree is based on 371 single-copy orthologs identified in all 79 genomes using OrthoFinder[26] v. 2.5.4. The genes were aligned using Mafft[44] v. 7.427 [--localpair --maxiterate 1000], concatenated, and filtered using TrimAL[45] v. 1.2 using -automated1 option. To determine the optimal substitution model and tree building, we used iqtree v. 1.6.12[46] [-m MFP -bb 1000 -alrt 1000 -safe]. For all downstream analyses, we used the Phylogenetic Hierarchical Orthogroups (HOGs) gene counts (Supplementary Data 3D) inferred on the root node of the species tree (N0), according to OrthoFinder guidelines (https://github.com/davidemms/OrthoFinder).

To infer ancestral ecological character states, each examined fungal lineage was assigned a character state based on their maximal/optimal growth rate: Thermophilic (≥45 °C), Thermotolerant (34 °C to 45 °C), and Mesophilic (≤34 °C). Ancestral ecological character states were reconstructed in Mesquite (http://www.mesquiteproject.org). The three ecological character states were analyzed using an unordered parsimony model, while binary ecological character states were analyzed using the Mk1 (Markov k-state 1 parameter model) likelihood reconstruction model (Supplementary Data 5).

We used Computational Analysis of Gene Family Evolution (CAFE)[11] v. 4.2.1 to study the gene expansions of the HOGs along with the evolutionary history. The Viterbi method was applied to them to compute branch-specific P-values to detect rapid expansions and contractions at each node. Only the gene families for which the Viterbi $P < 0.01$ were considered for further analysis.

The enrichment of Gene Ontology (GO)[47] terms was performed using a two-tailed Fisher's exact test (FET) and corrected for multiple testing using the Benjamini–Hochberg method in Python v.3.6 using the modules pandas v.1.1.5, numpy v.1.20.2 and scipy v.1.5.2. Mood's median nonparametric test[48] and Wilcoxon rank sum tests were used to evaluate the significance of GC3 differences between thermophilic, thermotolerant, and mesophilic fungi.

### Pangenome and codon usage analysis

We identified core, shell (common to two or more genomes), and unique genes based on OrthoFinder output (HOGs) for the pangenome analysis. The error bands (Fig. 2a, b) represent the standard deviation of 100 random subsamples from the study group. The Effective Number of Codons (ENC) and the third base of codon (GC3), which measures the degree of synonymous codon bias of gene or genome, were computed from coding sequences using CodonW v. 1.4.4[12].

### Machine learning to distinguish between thermophilic and mesophilic lifestyles

We used the Support Vector Machine method as implemented in Python's scikit-learn library (http://scikit-learn.org) to identify the most informative HOGs from OrthoFinder output for differentiating between thermophilic and thermophilic genomes, the same approach used by Haridas and collaborators[14]. The dataset was cleaned by removing HOGs with variance < 0.1 (Supplementary Data 3B). We used five subsets of thermophiles and mesophiles (20 and 30, respectively) (Supplementary Data 3B) to train and identify features (HOGs) that were differentially distributed between the pathogens and saprobes. Features were ranked on the ability to predict using C-Support Vector Classification within scikit-learn (sklearn.svm.SVC). The prediction strength was validated with five different training sets (Supplementary Data 3B) using sklearn.cross_validation and jackknife, and the best features were selected using sklearn.feature_selection.SelectKBest. Features highly correlated to lifestyle (>0.75, the top 10 HOGs in this analysis) were used in all combinations ("Combo" in Supplementary Data 3C) to improve prediction accuracy on the training set. The z-score for the heatmap (Fig. 4b) was calculated using the formula $z = (x-\mu)/\sigma$, the average and standard deviation calculated per HOGs.

### Protein structure prediction and clustering

The protein sequences from GH10 endoxylanases of the species used in this study were downloaded from MycoCosm[37], and signal peptide analysis was performed using signalP v. 4.1[49]. When detected, the signal peptide was removed from the amino acid sequence and then used for structural modeling. The structures of 318 endoxylanase sequences (Supplementary Data 4) were predicted by AlphaFold2[9] using collab-fold implementation[22]. Five models were generated for each protein, and we selected the best model (ranked_0.pdb) determined by the best average pLDDT score. Since the endoxylanase family shows high protein sequence similarity, we built a similarity network based on the structural similarity with all-vs-all comparison using FoldSeek[25] tmalign function with all PDB structure files predicted by AlphaFold2. We removed pairs with percent identity <0.7 and TMscore < 0.8 to obtain the structural-based clusters. The network was built and visualized using Prefuse Forced Directed Layout on Cytoscape v. 3.9.1[50].

To reconstruct the GH10 endoxylanase phylogeny, we used the amino acid sequences (same from AlphaFold2 predictions). The sequences were aligned using Mafft[44] v. 7.427 [--auto], concatenated, and filtered using TrimAL[45] v. 1.2 using -automated1 option. We used iqtree[46] v. 1.6.1218 [-m MFP -bb 10000 -safe] to determine the optimal substitution model and tree building (Supplementary Data 5).

### Reporting summary

Further information on research design is available in the Nature Portfolio Reporting Summary linked to this article.

## Data availability

Genome assemblies and annotations are available at MycoCosm (https://mycocosm.jgi.doe.gov) and deposited at DDBJ/ENA/GenBank under the following accessions: *Sodiomyces alcalophilus* JCM 7366 (JBBPEK000000000), *Thermothelomyces heterothallicus* CBS 203.75 (JBBPEJ000000000), *Thermothelomyces heterothallicus* CBS 202.75 (JBBPEI000000000), *Humicola hyalothermophila* CBS 454.80 (JBBEXK000000000), *Chaetomium olivicolor* CBS 102434 (JBBEXI000000000), *Corynascus sepedonium* ATCC 9787

(JBBEXH000000000), *Mycothermus thermophilus* CBS 625.91 (JBAGCS000000000), *Thermomyces stellatus* CBS 241.64 (JBAGCR000000000*), Phialemonium thermophilum* ATCC 24622 (JAZHXJ000000000), *Oculimacula yallundae* CBS 494.80 (JAZHXI000000000), *Thermothelomyces hinnuleus* ATCC 52474 (JAZ-GUK000000000), *Thermothelomyces myriococcoides* CBS 389.93 (JAZ-GUJ000000000), *Remersonia thermophila* ATCC 22073 (JAZGUE000000000), *Thermoascus aurantiacus* ATCC 26904 (JAZ-GUB000000000), *Thermomyces dupontii* NRRL 2155 (JAZGTC000000000), *Rasamsonia byssochlamydoides* NRRL 3658 (JAZGTB000000000), *Malbranchea cinnamomea* CBS 343.55 (JAZGTA000000000), *Thermothelomyces fergusii* CBS 405.69 (JAZGSZ000000000), *Mycothermus thermophilus* CBS 620.91 (JAZGSY000000000), *Corynascus similis* CBS 632.67 (JAZGRQ000000000), *Thermoascus thermophilus* ATCC 26413 (JAZGRP000000000), *Thermomyces lanuginosus* ATCC 200065 (JAZ-GRO000000000), *Rasamsonia emersonii* NRRL 3221 (JAZGRN000000000), *Sarocladium strictum* DS1bioAY4a (JAZGRM000000000), *Trametes versicolor* ATCC 20869 (JAZGQW000000000), *Rhizomucor pusillus* CBS 183.67 (JAZGQV000000000), *Calcarisporiella thermophila* CBS 279.70 (JAZGQU000000000), *Aureobasidium pullulans* ATCC 62921 (JAZBRX000000000), *Thermocarpiscus australiensis* ATCC 28236 (JAZBRW000000000), *Thermoascus crustaceus* CBS 181.67 (JAZBRV000000000). All GH10 protein structures in PDB format are available on OSF[51] - https://osf.io/cf569/.

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

## Acknowledgements
The work (proposals: https://doi.org/10.46936/10.25585/60000837 (thermophiles), https://doi.org/10.46936/10.25585/60007442 (*Thermoascus aurantiacus*), https://doi.org/10.46936/10.25585/60008172 (*Acremonium alcalophilum*), https://doi.org/10.46936/jejc.proj.2013.48100/60005302 (*Acremonium strictum*)) conducted by the U.S. Department of Energy Joint Genome Institute (https://ror.org/04xm1d337), a DOE Office of Science User Facility, is supported by the Office of Science of the U.S. Department of Energy operated under Contract No. DE-AC02-05CH11231. MVA-P. acknowledges funding by the Maria Zambrano Program from the Spanish Ministry of Science and Innovation (MICINN) for the attraction of international talent and grant PID2022-140187OB-I00 and by an UCOLIDERA grant from the University of Córdoba. AT acknowledges funding by Genome Canada and Genome Quebec for the support of the sequencing, assembly and annotation of the CSFG genomes.

## Author contributions
I.V.G., A.T., A.J.P., J.S., and D.O.N. designed the research and secured funding; M.-C.M., S.S., and A.J.R. sampled, cultured, and isolated the DNA/RNA for sequencing; A.S.S., B.A., K.L., A.K., S.M., R.R., R.O., S.H., A.L., J.G., A.C., K.D. and I.V.G. sequenced, assembled, and annotated the genomes; B.H. and E.D. predicted CAZyme genes in the analyzed genomes; M.V.A.-P., I.D.R. and T.T.M.N. curated genes and functional annotation. I.D.R., M.C.M., G.B., T.T.M.N., C.H., G.C., and M.I.H. performed investigation. A.S.S. and M.V.A.P. performed comparative analyses and bioinformatics; A.S.S., M.V.A.P., R.P.V., A.J.P., A.T., and I.V.G. wrote the paper. All authors read and commented on the manuscript.

## Competing interests
The authors declare no competing interests.

## Additional information

[1]US Department of Energy Joint Genome Institute, Lawrence Berkeley National Laboratory, Berkeley, CA, USA. [2]Centre for Structural and Functional Genomics, Concordia University, Montreal, QC, Canada. [3]Los Alamos National Laboratory, Los Alamos, NM, USA. [4]HudsonAlpha Institute for Biotechnology, Huntsville, AL, USA. [5]National Microbiome Data Collaborative, Lawrence Berkeley National Laboratory, Berkeley, CA, USA. [6]Department of Human Genetics, McGill University, Montreal, QC, Canada. [7]Bioinformatics Research Center, North Carolina State University, Raleigh, NC, USA. [8]Architecture et Fonction des Macromolécules Biologiques (AFMB), CNRS, Aix Marseille Université, Marseille, France. [9]DTU Bioengineering, Technical University of Denmark, 2800 Kgs, Lyngby, Denmark. [10]Woods Hole Oceanographic

Institution, Falmouth, MA, USA. [11]Joint BioEnergy Institute, Lawrence Berkeley National Laboratory, Berkeley, CA, USA. [12]Department of Biology, The University of New Mexico, Albuquerque, NM, USA. [13]Fungal Physiology, Westerdijk Fungal Biodiversity Institute & Fungal Molecular Physiology, Utrecht University, Utrecht, the Netherlands. [14]Systems Design and Architecture, Sandia National Laboratories, Albuquerque, NM 87123, USA. [15]Department of Plant and Microbial Biology, University of California Berkeley, Berkeley, CA, USA. [16]Present address: Departamento de Genética, University of Córdoba, 14071 Córdoba, Spain. ✉e-mail: ivgrigoriev@lbl.gov

