## [Peer Review file · Communications Biology]

Comparative Genomic Analysis of Thermophilic Fungi Reveals Convergent Evolutionary Adaptations and Gene Losses

Corresponding Author: Dr Igor Grigoriev

Version 0:

Reviewer comments:

Reviewer #1

(Remarks to the Author)

The authors of Insights into the molecular basis of fungal thermophily use a dataset of nearly 80 fungi to explore the underlying genetic contributions to thermophily. They identify several genetic patterns associated with thermophily, including gene content, assembly size, and codon content. They use both phylogenetic and machine learning methods to conduct their analyses.

The analysis identifies overall genomic patterns and specific genes that may be of interest to biotechnology. I have one specific major concern with the analysis and several smaller concerns that I believe should be addressed prior to publication.

Major Concern: The support vector classification analysis was trained on a subset of the thermophiles and mesophiles. Given the small number of species tested, the choice of the training data could greatly impact the results of the analysis. It would be an easy addition to iterate the training and testing sets to ensure that the specific training set is not biased or overfitting the results.

Minor Concerns

Lines 128-131 – Why was the Mk1 likelihood model used? Is there a logical justification for use of that specific model in mesquite?

Supp. File 1 - Supplemental file 1 should contain other important metadata such as assembly quality metrics (N50 or BUSCO), the ENC, and GC3 values. The N50 values are important for assessing the gene loss/gain data. The GC3 and ENC values would be necessary for re-analysis or future use of this data.

Phylogeny - Is a tree file provided? If not, it should be included. Does this tree have any conflicts with previously reported trees or is it consistent?

Ancestral Reconstruction – The authors should report the full output of Mesquite, including any rate parameters that were estimated.

Line 190-191 – The authors conclude that the thermophilic fungi have smaller genome sizes than their mesophilic counterparts. This is observational, and no statistics are provided. The authors should add a statistical test or they should temper the language in the conclusions (specifically lines 368-370)

Line 201 – What statistical test was used to generate this p-value. Was it the same test as the orthogroup analysis?

Line 199-207 – The authors report differences in CAZymes gene presence across groups. Given the significant clustering of the thermophilic fungi on the tree, the analysis would be more convincing if it was phylogenetically corrected. Or if gene presence/absence was shown to co-evolve with thermophilic yeasts.

Line 206-207 – The authors should provide a statistical test to justify the reduced number of GH

Figure 1 – The branch annotations should be defined. While it is logical to use red and green to represent the thermophilic and mesophilic fungi, it would be completely unreadable to individuals who are red/green colorblind.

Line 237 – The average gene gain and loss analysis is qualitative. The authors should conduct a statistical test to compare between the groups.

Line 237 – Another caveat to the gene gain/loss analysis is that it does not account for branch length difference. Does the total branch length differ between the two groups? If it does, then evolutionary rate or time could be contributing to the differences seen.

Figure 2a&b – The authors should define the shadow around the lines – is this a 95% confidence interval?

Line 259-260 – The authors should statistically test for a “fluctuation in ENC and GC3”.

Figure 3c – What statistical test was used here?

Figure 3d – The authors note that the thermophilic fungi are skewed towards the first quadrant. This is difficult to interpret because the points on the graph are overlapping and obscuring other points. The authors could divide the quadrants and

count the number of fungi in each quadrant. It also appears that there is a role for the group of fungi. Based on the way the data is presented, the Mucoromycota do not have the same pattern. It may also appear (although it is difficult to see) that the Basidiomycota are more likely to be in the 4th quadrant.

Line 278 – The authors should either report the code used to build the SVMs or provide the parameters used for each model constructed.

Line 288-289 – The analysis of all combinations of all clustered should be clarified. This is a significant portion of the results, and it is not entirely clear what “all combinations” mean. The authors should also report the full combinatorial results and not the top results.

Figure 4 – What is the z-score here and how was it calculated

Line 370 – Is “thermophilic” a form of specialization? Specialization would seem to be a reduction in temperature range. The designation of thermophilic fungi as specialists appears to be in connection with the “rarity” of high-temperature environments. Do these thermophilic fungi have reduced habitat niches? The authors should identify why they consider thermophilic fungi to be specialists.

Reviewer #2

(Remarks to the Author)

Steindorff et al have produced a new study looking at some of the genomic characteristics of thermophily across a diverse range of fungi. They have linked thermophily to several features of both the genomes and proteomes of these species.

In this reviewer's opinion, this work is novel and interesting. Moreover, I believe the stud's use of new protein modeling tools and structural similarity algorithms is timely and innovative.

My only reservations (detailed below) have to do with:

- 1) some aspects of data presentation are lacking (figure legends in particular) and may frustrate the reader
- 2) results of some analyses (particularly the AlphaFold results) need to be better related back to the bigger picture to contextualize when and where the numbers of disulfide bonds expand/contract.
- 3) the Methods section is wanting in many places. Some methods are not described at all.

Overall, I believe that if the authors can address these deficiencies, this paper will be a very nice addition to the literature on the evolutionary origins and molecular bases of thermophilic lifestyles in fungi.

===== MAJOR COMMENTS

Central to this study is the concept of an "optimal growth temperature", yet it is not clear how the optimal growth temperature for each species was established. Please include a description of this determination in the Methods as well as a table quantifying the growth results at all the temperatures tested (however they were measured). I suspect there is interspecific heterogeneity within each of these three phenotypic bins, and it would be useful to see how robust each of the assignments of "thermophily" is.

The Alphafold/structural approach (which I believe is both timely and well done) would greatly benefit if Figure 5 could relate the clusters back to the phylogeny in some way. Such a presentation would benefit overall story the paper is trying to tell, and could shed light on such questions as:

- + Do similar structures also group together on the tree?
- + When proteins of a cluster have low or high DS bond content, how does that bond content relate to neighboring taxa?
- + Where on the tree do those smaller clusters fall--close together or dispersed and are any at the ends of long branches?

The observation that disulfide bond content "may correlate with optimal high temperature" (lines 399-401) is tantalizing, and I would like to see the correlation tested in some way (e.g logistic regression or linear regression conditioned on of DS content). This would be helpful because I do not find such a correlation to be obvious when looking at the data as presented in Figure 5.

OTHER POINTS, QUESTIONS, and TYPOS

For the SVM, there were no "thermotolerant" species included in the Training set, yet the classifier invariably classifies ALL thermotolerant species as Mesophilic. The fact that thermotolerant species invariably "looks like" mesophilic species seems noteworthy to point out in the discussion.

The G3 differences are presented as significant, but this effect appears modest in the figure. Given that the statistical test used (Moods median) is not a common test, it would be helpful to also include the results from a more conventional non-parametric test such as Mann Whitney U test.

Supplemental file 3B (tab "SVM sets")

- + column F has incorrect heading
- + The confusion matrix is not 100% in correspondence to the SVM results reported (e.g. the Test Set Only matrix reports 1

TP and 1 FN whereas the table shows zero TP and 2 FN).

Line 287: should refer to table 3C

Figure 4

+ caption is very confusing, making several reference to the "four" informative clusters, when the figure and text seem to be referring to eight clusters

Line 339: list the percentage of GH10 xylanases to which temp optima could be assigned

Figure 5 needs to be better described to make it interpretable both on its own, as well as with how it's described in the main text. For example:

- + There are 3 structures presented in the figure--what their relationship to the proteins in the similarity network is not obvious.
- + What all the networks represent (structural clusters) needs to be spelled out.
- + The legend shows 3 outline colors for proteins assigned a temperature optima, yet there are clearly more than just three outline colors in the figure--what do these other colors represent? I believe these are relevant to the structures at the top--if so, simply drawing a line from the structure to the node might be more straightforward
- + It is noted that the proteins in Figure 5 fell into 2 orthogroups identified by Orthofinder (line 330)--are these orthogroups in evidence in the structural clusters? A sentence speaking to this question would be helpful.

It would be useful to include a one or two AlphaFold models colored by pLDDT (could be as part of Fig 5, could be in the Suppl files). Average pLDDT is a problematic metric, so seeing how confident the models are overall would add confidence to the Foldseek results.

To aid in reproducibility, please add the sequences used to generate each of the AlphaFold models to Suppl File 4

METHODS deficiencies:

- + optimal temperature determination needs to be described in detail
- + secondary metabolite cluster predictions is not described; SMURF is an older and little-used tool (these days) for BGC identification so describing the process is vital for
- + structural clustering methodology is not described in the methods at all

Reviewer #3

(Remarks to the Author)

Review of Steindorff et al for Communications Biology.

The manuscript describes multiple informatics approaches to understanding trends that distinguish thermophilic fungi from relatives adapted to lower temperatures. Specifically, orthogroup expansions and contractions, pangenomes, codon diversity, discriminating orthogroups, and protein structure networks were employed on a whole genome dataset including 79 isolates of different temperature preferences. Results were consistent with genome and codon diversity reduction in thermophiles, and a small number of PFAM domains that can predict thermophily with high confidence.

Comments:

38: change "a thermophile" to "thermophily" as this is a state.

132: capitalize CAFÉ as an acronym

134: what test results in the p-value? What does it signify here? A significant change?

154 and elsewhere: the term "cluster" is used ambiguously. Distinguish among orthogroups, hierarchical clusters, etc (even if only one thing that could be called a "cluster" is present).

188: "variation (in) genome features"

196-198: rewrite sentence to logic of the paragraph. Better yet, this should be moved to discussion as it would probably still be out of place when describing genome sizes in Pezizomycotina.

199-206: Should a phylogenetic correction be applied to these tests? It is not clear how the CAFÉ analysis is being used here.

223: Eurotiomycetes class

Figure 4 introduces a lot of confusion around the number of PFAMs and Orthogroups that are involved in these analyses.

What has 4 and what has 8, and how do the Orthogroups relate to the PFAMs? I think there might be some errors in the mix. B says 4 orthogroups but shows 8 PFAMs, for instance.

337 last clause: inference, to discussion?

Main concern: How were the sequences of the proteins in the analysis in Figure 5 validated? RNAseq? Manual curation of alignments? This is important because de novo/outgroup referenced annotation can introduce artifacts. This might even be more prevalent in a species with GC bias that might change prevalence of splice variants.

358: change "returning" to "dating"

368-377: it seems dimorphism and dual-lifestyles should be discussed as a confounder in this context.

376: found in deep inner cores of compost piles (which are hot)? Certainly UV protection is important to molds on surfaces of composts.

382: remove unnecessary comma

391-393: please explain why, with regards to DNA melting temperature in a tradeoff with N-limitation.

Author Rebuttal letter:

1

We would like to thank the Editor and all Reviewers for their careful reading of our manuscript and insightful comments. We made significant revisions to address most of the reviewers' concerns and believe we substantially improved the revised manuscript. Below, you can find the answers written in red. Also, the modifications in the manuscript are also shown in red.

Reviewer #1 (Remarks to the Author):

The authors of Insights into the molecular basis of fungal thermophily use a dataset of nearly 80 fungi to explore the underlying genetic contributions to thermophily. They identify several genetic patterns associated with thermophily, including gene content, assembly size, and codon content. They use both phylogenetic and machine learning methods to conduct their analyses. The analysis identifies overall genomic patterns and specific genes that may be of interest to biotechnology. I have one specific major concern with the analysis and several smaller concerns that I believe should be addressed prior to publication.

Major Concern: The support vector classification analysis was trained on a subset of the thermophiles and mesophiles. Given the small number of species tested, the choice of the training data could greatly impact the results of the analysis. It would be an easy addition to iterate the training and testing sets to ensure that the specific training set is not biased or overfitting the results.

That is a relevant concern. We re-ran our SVM classifier with 5 different training sets and added the average/standard deviation of the SVM score. We found very similar results, with the eight HOGs ranked top scores. To give more confidence to the analysis, we used the Wilcoxon rank sum test to assign a p-value and FDR to each cluster. Only one of the HOGs was not significant based on the Wilcoxon test, but we decided to keep it because it has the second-highest SVM score. All new data is available in the supplemental file 3.

Minor Concerns

Lines 128-131 â Why was the Mk1 likelihood model used? Is there a logical justification for use of that specific model in mesquite?

According to the Mesquite manual, they offer two options for the ancestral state likelihood model (<https://www.mesquiteproject.org/Ancestral%20States.html#likelihood>): Mk1 and AsymmMk model. The latter supports only binary (0,1) characters. In our case, we used three characters (thermophilic, thermotolerant, and mesophilic).

Supp. File 1 - Supplemental file 1 should contain other important metadata such as assembly quality metrics (N50 or BUSCO), the ENC, and GC3 values. The N50 values are important for

assessing the gene loss/gain data. The GC3 and ENC values would be necessary for re-analysis or future use of this data.

Thank you for the suggestion. BUSCO, N50/L50, ENC, and GC3 were added to the Supplemental File 1.

Phylogeny - Is a tree file provided? If not, it should be included. Does this tree have any conflicts with previously reported trees or is it consistent?

The tree file is provided in supplemental file 4. The phylogeny matches the NCBI taxonomy levels and multi-locus tree from Morgenstern et al., 2012 (<https://doi.org/10.1016/j.funbio.2012.01.010>).

Ancestral Reconstruction â The authors should report the full output of Mesquite, including any rate parameters that were estimated.

The full output from Mesquite is now provided as supplemental file 4.

Line 190-191 â The authors conclude that the thermophilic fungi have smaller genome sizes

than their mesophilic counterparts. This is observational, and no statistics are provided. The authors should add a statistical test or they should temper the language in the conclusions (specifically lines 368-370)

A P-value was added for all thermophilic/mesophilic genome sizes using the Wilcoxon rank sum exact test. The other examples are 1x1 comparisons or 2x1 for Thermomyces. For this reason, we kept the percentage difference.

Line 201 â What statistical test was used to generate this p-value. Was it the same test as the orthogroup analysis?

It is a Wilcoxon rank sum exact test. Added to the text.

Line 199-207 â The authors report differences in CAZymes gene presence across groups. Given the significant clustering of the thermophilic fungi on the tree, the analysis would be more convincing if it was phylogenetically corrected. Or if gene presence/absence was shown to co-evolve with thermophilic yeasts.

We have partitioned the analysis into the clades Chaetomiaceae and Eurotiomycetes (which have a larger presence of thermophilic fungi) to separate by phylogenetic clades. Grouping them together diluted the signal, with no clear coevolution of CAZymes presence/absence.

Line 206-207 â The authors should provide a statistical test to justify the reduced number of GH A p-value was added to show the statistical difference between mesophilic and human pathogens.

Figure 1 â The branch annotations should be defined. While it is logical to use red and green to represent the thermophilic and mesophilic fungi, it would be completely unreadable to individuals who are red/green colorblind.

For all figures, we changed the red/green to a more colorblind-friendly version of those colors (<https://visualisingdata.com/2019/08/five-ways-to-design-for-red-green-colour-blindness/>).

Line 237 â The average gene gain and loss analysis is qualitative. The authors should conduct a statistical test to compare between the groups.

Added a p-value for the average differences mentioned in the text.

Line 237 â Another caveat to the gene gain/loss analysis is that it does not account for branch
3

length difference. Does the total branch length differ between the two groups? If it does, then evolutionary rate or time could be contributing to the differences seen.

Thank you for pointing that out. The CAFE software, indeed, does not consider the evolutionary rate, only tree topology. However, the text focused on the interpretation at the family level nodes (Chaetomiaceae, Thermoascaceae, Trichocoaceae). At this level, the family trees have close evolutionary rates calculated using phykit software [phykit evolutionary_rate] - <https://jsteenwyk.com/PhyKIT/usage/index.html#evolutionary-rate>, namely 0.059, 0.08, and 0.092 for Chaetomiaceae, Thermoascaceae, and Trichocoaceae, respectively.

Only Chaetomiaceae has a lower evolutionary rate, but this is because we have more than 3 times more genomes from this family. Therefore, the branch length is irrelevant in the context in which we performed this analysis.

Figure 2a&b â The authors should define the shadow around the lines â is this a 95% confidence interval

The shadow represents the standard deviation of each average point. Added to the legend.

Line 259-260 â The authors should statistically test for a fluctuation in ENC and GC3â.

The Pearson correlation and p-value for Basidiomycota and Mucoromycota were added and compared to Ascomycota, showing only the significant differences for the latter.

Figure 3c â What statistical test was used here?

Here, we used the Wilcoxon rank sum nonparametric test. The p-value for Chaetomiaceae was missing, but both were included in the text.

Figure 3d â The authors note that the thermophilic fungi are skewed towards the first quadrant. This is difficult to interpret because the points on the graph are overlapping and obscuring other points. The authors could divide the quadrants and count the number of fungi in each quadrant. It also appears that there is a role for the group of fungi. Based on the way the data is presented, the Mucoromycota do not have the same pattern. It may also appear (although it is difficult to see) that the Basidiomycota are more likely to be in the 4th quadrant.

Thank you for the suggestion. We added a dashed line separating the quadrants based on the average GC3 and ENC. Indeed, all Mucoromycota are present in the 4th quadrant. We added to

the text explaining this.

Line 278 – The authors should either report the code used to build the SVMs or provide the parameters used for each model constructed.

The parameters are reported in the methods section. Some extra information was included.

Line 288-289 – The analysis of all combinations of all clustered should be clarified. This is a significant portion of the results, and it is not entirely clear what “all combinations” mean. The authors should also report the full combinatorial results and not the top results.

The combinatorial analysis was performed with the top ranked scores and with > 0.75 correlation with the mesophile/thermophile split. All the results for the ranks and combinatorial analysis are presented in Supplemental File 3.

Figure 4 – What is the z-score here and how was it calculated

The following sentence was added to the methods section: The z-score for the heatmap (Figure 4b) was calculated using the formula $z = (x - \bar{x})/\sqrt{s^2}$, the average and standard deviation calculated per OrthoFinder cluster.

Line 370 – Is “thermophilic” a form of specialization? Specialization would seem to be a

4

reduction in temperature range. The designation of thermophilic fungi as specialists appears to be in connection with the “rarity” of high-temperature environments. Do these thermophilic fungi have reduced habitat niches? The authors should identify why they consider thermophilic fungi to be specialists.

We revised the second paragraph of the discussion to address the question of specialization and connect that issue to the evolution of thermophily and its possible consequences for niche space. Although we consider thermophily to reflect a type of specialization, the current wording hedges our claims on this issue, while at the same time it addresses why high-temperature environments can lead to specialization among organisms that are adapted to thrive in them.

Reviewer #2 (Remarks to the Author):

Steindorff et al have produced a new study looking at some of the genomic characteristics of thermophily across a diverse range of fungi. They have linked thermophily to several features of both the genomes and proteomes of these species.

In this reviewer’s opinion, this work is novel and interesting. Moreover, I believe the study’s use of new protein modeling tools and structural similarity algorithms is timely and innovative.

My only reservations (detailed below) have to do with:

- 1) some aspects of data presentation are lacking (figure legends in particular) and may frustrate the reader
- 2) results of some analyses (particularly the AlphaFold results) need to be better related back to the bigger picture to contextualize when and where the numbers of disulfide bonds expand/contract.
- 3) the Methods section is wanting in many places. Some methods are not described at all.

Overall, I believe that if the authors can address these deficiencies, this paper will be a very nice addition to the literature on the evolutionary origins and molecular bases of thermophilic lifestyles in fungi.

=====
MAJOR COMMENTS

Central to this study is the concept of an “optimal growth temperature”, yet it is not clear how the optimal growth temperature for each species was established. Please include a description of this determination in the Methods as well as a table quantifying the growth results at all the temperatures tested (however they were measured). I suspect there is interspecific heterogeneity within each of these three phenotypic bins, and it would be useful to see how robust each of the assignments of “thermophily” is.

In terms of heterogeneity, substantial evidence points to both the within-species consistency of thermophily and the general rarity of the thermophile lifestyle among fungi. While a degree of heterogeneity in optimal growth temperature exists across species and across orders (see for example Powell et al. 2012, *Mycologia* 104:813-825), we know of no examples where both thermophiles and mesophiles exist within a single fungal species or even within a lineage of very closely-related species. Quite the contrary, dating back to the earliest studies of

thermophilic fungi [Rhizomucor pusillus (aka Mucor pusillus), described by Lindt in 1886 (Arch. Exp. Path. Pharmacol.: 272); Thermomyces lanuginosus, described by Tsiklinskaya in 1899 (Ann. Inst. Pasteur 13: 500); the reporting of these and other species by Miede in 1907 (Die Selbsterhitzung des heus: eine biologische Studie, Gustav Fischer, Jena); and the

5

characterization of these and new thermophilic species by Cooney and Emerson in their 1964 monograph (Thermophilic Fungi; an account of Their Biology, Activities, and Classification, 1964, W.H. Freeman)], many of these fungi and others that are the subject of the current study have been reported from diverse efforts designed to isolate thermophiles. In contrast, to our knowledge none of these species have been reported to have mesophilic members. And while it is clear that thermophily has been gained and lost independently in certain groups of Ascomycota, the acquisition of thermophily has been extremely rare, as evidenced by the fact that only a tiny fraction of Ascomycota and Mucoromycota have evolved to be thermophilic. We better described the optimal temperature in the methods (lines 106 - 110). Also, the Morgenstern et al., 2012, et al. citation, since all the thermophilic/thermotolerant strains originated from the same lab.

The AlphaFold/structural approach (which I believe is both timely and well done) would greatly benefit if Figure 5 could relate the clusters back to the phylogeny in some way. Such a presentation would benefit overall story the paper is trying to tell, and could shed light on such questions as:

Thank you for the suggestion. We've built a phylogenetic tree with all GH10 and added it to Figure 5. Hopefully, it will help answer the question.

+ Do similar structures also group together on the tree?

Similar structures often cluster together, but not always. We found that the disulfide bonds hold the structure together, with examples of high branch length and lower number of disulfide numbers not present in the same structural cluster.

+ When proteins of a cluster have low or high DS bond content, how does that bond content relate to neighboring taxa?

Yes, the high and low DS bond content tend to cluster together as now shown on the black outer ring of Fig 5a.

+ Where on the tree do those smaller clusters fall--close together or dispersed and are any at the ends of long branches?

We found that smaller clusters (< 7 proteins) tend to cluster together, forming clade-specific (i.e., Aspergillus/Penicillium) groups. And singletons are dispersed along the tree.

The observation that disulfide bond content "may correlate with optimal high temperature" (lines 399-401) is tantalizing, and I would like to see the correlation tested in some way (e.g logistic regression or linear regression conditioned on of DS content). This would be helpful because I do not find such a correlation to be obvious when looking at the data as presented in Figure 5. Unfortunately, we don't have enough points for a meaningful correlation of optimal temperature and DS bonds. Most endo xylanases for which we have optimal temperatures fall in the same cluster. We decided to leave it as a "may" because we see 9 endoxyanases with optimal temperature ~60°C and 2 DS bonds, but we can't extrapolate because others with lower optimal temperature also have 2 DS bonds. Also, optimal temperature assays are variable depending on the lab and methods used. We think it's better to leave it as an idea for future work.

OTHER POINTS, QUESTIONS, and TYPOS

For the SVM, there were no "thermotolerant" species included in the Training set, yet the classifier invariably classifies ALL thermotolerant species as Mesophilic. The fact that

6

thermotolerant species invariably "looks like" mesophilic species seems noteworthy to point out in the discussion.

Thanks for the suggestion. We added a sentence in the results that mentioned this fact.

The G3 differences are presented as significant, but this effect appears modest in the figure. Given that the statistical test used (Moods median) is not a common test, it would be helpful to also include the results from a more conventional non-parametric test such as Mann Whitney U test.

We included both Moods median and Wilcoxon rank sum test (aka Mann Whitney U test) in the figure.

Supplemental file 3B (tab "SVM sets")
+ column F has incorrect heading
Fixed.

+ The confusion matrix is not 100% in correspondence to the SVM results reported (e.g. the Test Set Only matrix reports 1 TP and 1 FN whereas the table shows zero TP and 2 FN). Thanks for finding this error. We fixed the confusion table in the supplemental file 3.

Line 287: should refer to table 3C
Fixed.

Figure 4

+ caption is very confusing, making several reference to the "four" informative clusters, when the figure and text seem to be referring to eight clusters
That is a mistake, we replaced with eight.

Line 339: list the percentage of GH10 xylanases to which temp optima could be assigned
Number and percentage added to the text.

Figure 5 needs to be better described to make it interpretable both on its own, as well as with how it's described in the main text. For example:

+ There are 3 structures presented in the figure--what their relationship to the proteins in the similarity network is not obvious.

These three structures were added to illustrate previously published endoxylanases. Since it was requested to add the structures colored by pLDDT, we removed these three and left only the new structures in Figure 5c.

+ What all the networks represent (structural clusters) needs to be spelled out.
Information about the network has been added to the Figure 5 legend.

+ The legend shows 3 outline colors for proteins assigned a temperature optima, yet there are clearly more than just three outline colors in the figure--what do these other colors represent? I believe these are relevant to the structures at the top--if so, simply drawing a line from the structure to the node might be more straightforward

As explained before, it was mostly to show some examples. They were removed and replaced by 9 examples of structures colored by pLDDT (Figure 5c).

+ It is noted that the proteins in Figure 5 fell into 2 orthogroups identified by Orthofinder (line 7

330)--are these orthogroups in evidence in the structural clusters? A sentence speaking to this question would be helpful.

Yes, the smaller Orthofinder cluster (HOG) is in the root (and cluster 9 of Figure 5a and b) of the phylogenetic tree. A clarifying sentence was added.

It would be useful to include a one or two AlphaFold models colored by pLDDT (could be as part of Fig 5, could be in the Suppl files). Average pLDDT is a problematic metric, so seeing how confident the models are overall would add confidence to the Foldseek results.

Thanks for the suggestion. We randomly selected one protein structure per cluster the structure based on the pLDDT and included it in figure 5.

To aid in reproducibility, please add the sequences used to generate each of the AlphaFold models to Suppl File 4
Sequences added to Supplemental file 4.

METHODS deficiencies:

+ optimal temperature determination needs to be described in detail
We better described the optimal temperature in the methods (lines 106 - 110).

+ secondary metabolite cluster predictions is not described; SMURF is an older and little-used tool (these days) for BGC identification so describing the process is vital for
Even though SMURF was discontinued years ago, it is a fungal-specific tool. In our internal tests, SMURF results correlate well with predictions generated by antiSmash. For a more accurate comparison, we decided to use our version of SMURF with the respective families.

+ structural clustering methodology is not described in the methods at all
Please check the section "Protein structure prediction and clustering" describing structural

clustering in detail.

Reviewer #3 (Remarks to the Author):

Review of Steindorff et al for Communications Biology.

The manuscript describes multiple informatics approaches to understanding trends that distinguish thermophilic fungi from relatives adapted to lower temperatures. Specifically, orthogroup expansions and contractions, pangenomes, codon diversity, discriminating orthogroups, and protein structure networks were employed on a whole genome dataset including 79 isolates of different temperature preferences. Results were consistent with genome and codon diversity reduction in thermophiles, and a small number of PFAM domains that can predict thermophily with high confidence.

8

Comments:

38: change "thermophile" to "thermophily" as this is a state.

Fixed.

132: capitalize CAFÉ as an acronym

Fixed.

134: what test results in the p-value? What does it signify here? A significant change?
It's the Viterbi method applied by CAFE. We inverted the sentences to clarify.

154 and elsewhere: the term "cluster" is used ambiguously. Distinguish among orthogroups, hierarchical clusters, etc (even if only one thing that could be called a "cluster" is present). To avoid confusion, we replaced all the cluster/orthogroup words with HOGs (Phylogenetic Hierarchical orthogroups), and added a sentence explaining this in the methods.

188: "variation (in) genome features"

Fixed.

196-198: rewrite sentence to logic of the paragraph. Better yet, this should be moved to discussion as it would probably still be out of place when describing genome sizes in Pezizomycotina.

The corresponding sentence has been moved to the discussion section.

199-206: Should a phylogenetic correction be applied to these tests? It is not clear how the CAFÉ analysis is being used here.

As mentioned previously, CAFE, indeed, does not consider the evolutionary rate, only tree topology. However, the text focused on the interpretation at the family level nodes (Chaetomiaceae, Thermoascaceae, Trichocoaceae). At this level, the family trees have close evolutionary rates calculated using phykit software. Only Chaetomiaceae has a lower evolutionary rate, but this is because we have more than 3 times more genomes from this family. Therefore, the branch length is irrelevant in the context in which we performed this analysis.

223: Eurotiomycetes class

Fixed.

Figure 4 introduces a lot of confusion around the number of PFAMs and Orthogroups that are involved in these analyses. What has 4 and what has 8, and how do the Orthogroups relate to the PFAMs? I think there might be some errors in the mix. B says 4 orthogroups but shows 8 PFAMs, for instance.

Thank you for this observation. We used PFAM in the figure to emphasize the function since the majority (>70%) of the proteins in this cluster contain that PFAM domain. The "four" was a mistake; we replaced it with "eight."

337 last clause: inference, to discussion?

Thank you for the suggestion. We moved that inference to the discussion section.

Main concern: How were the sequences of the proteins in the analysis in Figure 5 validated? RNAseq? Manual curation of alignments? This is important because de novo/outgroup referenced annotation can introduce artifacts. This might even be more prevalent in a species with GC bias that might change prevalence of splice variants.

9

All genomes, except for six of the previously published genomes (Stalo1, Chale1, Corin1, Corno1, Chafun1, Chafu1), used RNAseq data to support gene predictions. For CAZy annotation, module composition and family assignment of all carbohydrate-active enzymes were performed just as for the daily updates of the CAZy database (<http://www.cazy.org>). Briefly, the method combines BLAST and HMMer searches conducted against sequence libraries and HMM profiles made of the individual functional modules featured in the CAZy database. All positive hits were manually examined by human curators for final validation.

358: change "returning" to "dating"
Changed.

368-377: it seems dimorphism and dual-lifestyles should be discussed as a confounder in this context.

In general, dimorphic fungi and dual lifestyle fungi require some temperature-derived signals, including several transcription factors, to trigger dimorphic switching, which is not the case for thermophilic fungi. Based on our experience, dimorphism is not necessarily related to temperature, as we also see carbon source-dependent and sexual state-dependent differences in yeast/filamentous form. We believe such a comparison is not the focus of this study, but it is a great idea for future investigation.

376: found in deep inner cores of compost piles (which are hot)? Certainly UV protection is important to molds on surfaces of composts.

Yes, they are usually found inside compost piles in places that reach higher temperatures.

382: remove unnecessary comma
Removed.

391-393: please explain why, with regards to DNA melting temperature in a tradeoff with N-limitation.

Since thermophilic fungi inhabit high-temperature environments, an increase in GC3 is associated with specific codon patterns (we could not find significant differences), which can enhance translation efficiency and DNA and protein stability. Therefore, this increase can be an adaptation to manage nitrogen constraints.

We extended the sentence explaining possible impacts, but since we have not discussed nitrogen limitation, we decided not to add this part.

Version 1:

Reviewer comments:

Reviewer #1

(Remarks to the Author)

The authors of Insights into the molecular basis of fungal thermophily use a dataset of nearly 80 fungi to explore the underlying genetic contributions to thermophily. They have sufficiently addressed the reviewer's comments and have substantially improved the manuscript in doing so.

I have no further comments at this time. The manuscript contributes a novel analysis to the study of thermophily in fungi.

Reviewer #2

(Remarks to the Author)

I am sufficiently satisfied with the revised version of this manuscript.

Reviewer #3

(Remarks to the Author)

my concerns were adequately addressed
